# Deep Learning in Controlled Environment Agriculture: A Review of Recent Advancements, Challenges and Prospects

**DOI:** 10.3390/s22207965

**Published:** 2022-10-19

**Authors:** Mike O. Ojo, Azlan Zahid

**Affiliations:** Department of Biological and Agricultural Engineering, Texas A&M AgriLife Research, Texas A&M University System, Dallas, TX 75252, USA

**Keywords:** smart farming, greenhouse, deep neural networks, indoor agriculture, plant factory, protected agriculture, vertical farm, smart agriculture, deep learning

## Abstract

Controlled environment agriculture (CEA) is an unconventional production system that is resource efficient, uses less space, and produces higher yields. Deep learning (DL) has recently been introduced in CEA for different applications including crop monitoring, detecting biotic and abiotic stresses, irrigation, microclimate prediction, energy efficient controls, and crop growth prediction. However, no review study assess DL’s state of the art to solve diverse problems in CEA. To fill this gap, we systematically reviewed DL methods applied to CEA. The review framework was established by following a series of inclusion and exclusion criteria. After extensive screening, we reviewed a total of 72 studies to extract the useful information. The key contributions of this article are the following: an overview of DL applications in different CEA facilities, including greenhouse, plant factory, and vertical farm, is presented. We found that majority of the studies are focused on DL applications in greenhouses (82%), with the primary application as yield estimation (31%) and growth monitoring (21%). We also analyzed commonly used DL models, evaluation parameters, and optimizers in CEA production. From the analysis, we found that convolutional neural network (CNN) is the most widely used DL model (79%), Adaptive Moment Estimation (Adam) is the widely used optimizer (53%), and accuracy is the widely used evaluation parameter (21%). Interestingly, all studies focused on DL for the microclimate of CEA used RMSE as a model evaluation parameter. In the end, we also discussed the current challenges and future research directions in this domain.

## 1. Introduction

Sustainable access to high-quality food is a problem in developed and developing countries. Rapid urbanization, climate change, and depleting natural resources have raised the concern for global food security. Additionally, the rapid population growth further aggregate the food insecurity challenge. According to World Health Organization, the food production needs to be increased by 70% to meet the food demand of about 10 billion people by 2050 [1], of which about 6.5 billion will be living in urban areas [2]. A significant amount of food is produced in the open fields using traditional agricultural practices, which results in low yields per sq. ft of land used. Simply increasing the agricultural land is not a long-term option because of the associated risks of land degradation, de-forestation, and increased emissions due to transportation to urban areas [3]. Thus, alternative production systems are essential to offset these challenges for establishing a sustainable food supply chain.

Controlled environment agriculture (CEA), including greenhouses, high-tunnels, vertical farms (vertical or horizontal plane), and plant factories, is increasingly considered an important strategy to address global food challenges [4]. CEA is further categorized based on the growing medium and production technology (hydroponics, aquaponics, aeroponics, and soil-based). CEA integrates knowledge across multiple disciplines to optimize crop quality and production efficiency without sufficient arable land. Globally, the CEA market has witnessed a growth of about 19% in 2020 and is projected to grow at a compound annual growth rate of 25% during the 2021–28 period [5]. CEA market in the US is predicted to be $3 billion by 2024, with an annual growth of about 24% [6]. Advocates of CEA claim that the system is more than 90% efficient in water use, produces 10–250 times the higher yield per unit area, and generates 80% less waste than traditional field production, while also reducing food transportation miles in urban areas [3,7,8].

Despite all these benefits, the CEA industry struggles to achieve economic sustainability due to inefficient microclimate and rootzone-environment controls and high costs. Microclimate control, including light, temperature, airflow, carbon dioxide, and humidity, is a major challenge in CEA, which is essential to produce uniform, high quantity, and quality crops [9]. In the last decade, substantial research has been carried out on implementing intelligent systems in CEA facilities such as nutrient solution management for hydroponic farm [10], and cloud-based micro-environment monitoring and control systems for the vertical farm [11]. Further, using artificial intelligence (AI) algorithms have also created new opportunities for intelligent predictions and self-learning [12]. DL has gained significant attention in the last few years due to its massive footprints in many modern day technologies. DL algorithms applied to CEA across all units have provided insights into farmers’ support and action. Computer vision and DL algorithms have been implemented to automate the irrigation in vertical stack farms [13], and microclimate control [14], which facilitated the growers to carry out a quantitative assessment for high-level decision-making.

CEA is an intensive production system, the labor is required year-round, and the labor requirement is also significantly higher than traditional agriculture [15]. A small indoor farm of fewer than 1500 sq. ft requires at least three full-time workers [16]. Intelligent automation, however, could address these challenges. Furthermore, the crop cycle in CEA is relatively small, therefore the timely decision to perform a specific operation is critical. For instance, the harvest decision requires information about crop maturity, which can be obtained using an optical sensor integrated with DL-based prediction models [17]. In recent years, research has been carried out to develop robotic systems for indoor agriculture [18,19,20]. For target detection, various sensors are implemented such as cameras [19], or LiDAR [21]. This increasing popularity of DL applications in CEA sparks our motivation to conduct a systematic review of recent advances in this domain.

### 1.1. Review Scope

Table 1 presents the existing review articles covering DL applications in different sections of agriculture [22,23,24,25,26,27,28]. From the table, it is evident that the reported studies (based on the authors’ knowledge) lacks a critical overview of recent advancements in DL methodologies for CEA. Thus, a need to review the recent works in CEA is consequential to determine state of the art, identify current challenges, and provide future recommendations. Figure 1 shows the bibliometric network and co-occurrence map of the author-supplied keywords.

### 1.2. Paper Organization

The article’s organization is as follows: Section 2 features the methodology of the review process, including establishing review protocol, keywords selection, research questions formation, and data extraction. Section 3 presents the results of the review, including data synthesis and answers to the core research questions. Existing challenges and future recommendations are discussed in Section 4. The overall conclusions of the review is presented in Section 5.

## 2. Research Methodology

### 2.1. Review Protocol

In this research, we adhered to the SLR standard approach as described by Chitu Okoli and Kira Schabram [29]. Using this approach, we identified, specified, and analyzed all the publications in DL for CEA applications from 2019 to date, in order to present a response to each research question (RQ) and identify any gaps. Planning, conducting, and reporting the review are the three parts we divided the SLR process into. Figure 2 depicts the actions taken at each level of the SLR. During the planning phase we identified RQs, relevant keywords, and databases. After the RQs were prepared, the search protocol was created, along with which databases and search strings should be used. Search string for each database was generated using selected keywords. Wiley, Web of Science, IEEEXplore Springer Link, Google Scholar, Scopus, and Science Direct are the databases used in this study. The databases were chosen to ensure adequate coverage of the target sector and to increase the scope of the assessment. By going through all the eligible studies, pertinent studies were chosen for the conducting review stage. Significant information was retrieved from the publications that met the selection/inclusion criteria in response to the RQs. Extracted data from selected publications were used to answer the RQs during the reporting stage, and the outcomes were presented using accompanying visuals and summary tables. This type of literature analysis demonstrates the most recent findings of DL research in CEA.

### 2.2. Research Questions

Identifying RQs is essential to the systematic review. At the start of the study, we set the RQs up to adhere to the review procedure. The searched articles were examined from a variety of aspects, and the following RQs were established.

RQ.1: What are the most often utilized DL models in CEA, and their benefits and drawbacks?RQ.2: What are the main application domains of DL in CEA?RQ.3: What evaluation parameters are used for DL models in CEA?RQ.4: What are the DL backbone networks used in CEA applications?RQ.5: What are the optimization methods used for CEA applications?RQ.6: What are the primary growing media and plants used for DL in the CEA?

### 2.3. Search Method

In order to focus the search results on papers that were specifically relevant to the SLR’s scope, a methodical approach was taken. The original search was conducted using a generalized search equation that included the necessary keywords “deep learning” AND “controlled environment agriculture” OR “greenhouse” OR “plant factory” OR “vertical farm” to obtain the expanded search results. From the search results, a few studies were selected to extract the author supplied keywords, and synonyms. The discovered keywords produced the general search string/equation: (“controlled environment agriculture” OR “greenhouse” OR “plant factory” OR “vertical farm” OR “indoor farm”) AND (“deep learning” OR “deep neural network”). All seven databases were searched using the same keywords. Following search strings were used for different databases:**Science Direct:** (“controlled environment agriculture” OR “greenhouse” OR “plant factory” OR “vertical farm”) AND (“Deep Learning”) NOT (“Internet of Things” OR “GREENHOUSE GAS” OR “gas emissions” OR “Machine learning”)**Wiley:** (“controlled environment agriculture” OR “greenhouse” OR “plant factory” OR “vertical farm*”) AND (“deep learning”) NOT (“Internet of Things” OR “greenhouse gas” OR “Gas emissions” OR “machine learning” OR “Review”)**Web of Science:** (AB = (((“controlled environment agriculture” OR “vertical farm” OR “greenhouse” OR “plant factory”) AND (“deep learning” ) NOT ( “Gas Emissions” OR “Internet of Things” OR “Greenhouse Gas” OR “machine learning” OR “Review”))))**Springer Link:** (“deep learning”) AND (“Greenhouse” OR “controlled environment agriculture” OR “vertical farm” OR “plant factory”) NOT (“Internet of things” OR “review” OR “survey” OR “greenhouse gas” OR “IoT” OR “machine learning” OR “gas emissions”)**Google Scholar:** “greenhouse” OR “vertical farm” OR “controlled environment agriculture” OR “plant factory” “deep learning”—“Internet of Things”—“IoT”—“greenhouse gas”—“review”—“survey”—“greenhouse gases”—“Gas Emissions”—“machine learning”**Scopus:** TITLE-ABS-KEY ((“deep learning”) AND (“vertical farm*” OR “controlled environment agriculture” OR “plant factory” OR “greenhouse”)) AND (LIMIT-TO (PUBYEAR, 2022 ) OR LIMIT-TO (PUBYEAR, 2021) OR LIMIT-TO ( PUBYEAR, 2020) OR LIMIT-TO ( PUBYEAR, 2019 )) AND (LIMIT-TO (LANGUAGE, “English” )) AND (EXCLUDE (EXACTKEYWORD, “Greenhouse Gases”) OR EXCLUDE ( EXACTKEYWORD, “Gas Emissions”) OR EXCLUDE (EXACTKEYWORD, “Machine Learning”) OR EXCLUDE (EXACTKEYWORD, “Internet of Things”))**IEEEXplore:** (“controlled environment agriculture” OR “greenhouse” OR “plant factory” OR “vertical farm”) AND (“Deep Learning”) NOT (“Internet of Things” OR “GREENHOUSE GAS” OR “gas emissions” OR “Machine learning”)

After all the results were processed, a total of 751 studies were found using the aforementioned search strings.

### 2.4. Selection/Inclusion Criteria

To establish the limits for the SLR, the inclusion Criteria (IC) and exclusion Criteria (EC) were defined. To choose the pertinent research based on the IC and EC, the studies that were obtained from all databases were carefully examined. The search outcomes from several databases were combined in a spreadsheet and compared to all of the IC and EC. A study must meet all of the ICs and ECs in order to be considered for the review. Upon passing the IC and EC, all studies that could respond to the RQs were deemed pertinent and chosen. The ICs and ECs are presented below:IC.1: Peer-reviewed journal publications and conference papers.IC.2: Studies published during the period between 2019 and April 2022.IC.3: Studies should offer answers to the RQs.EC.1: Study unrelated to DL for CEA.EC.2: Full text not accessible.EC.3: Duplicate or obtained from another database.EC.4: Publication is a review or survey article.EC.5: Publications such as book reviews, editorials, and summaries of conferences and seminars are not subjected to peer review.EC.6: Studies published before 2019.

Applying the ICs and ECs produced a total of 72 eligible articles were selected, which were then shortlisted for additional examination. An overview of article search and selection procedure is shown Figure 3. The distribution of selected papers from different databases is shown in Table 2.

### 2.5. Data Extraction

Table 3 and Table 4 presents the summary of studies that fulfilled the selection criteria. The necessary data required to answer the RQs, were extracted from the selected studies. The extracted data were summarized using a spreadsheet application. In the spreadsheet, each study was assigned to separate row, and the column was assigned to different parameters. Tasks, DL model, training networks, imaging system, optimizer, pre-processing augmentation, application domain, performance parameters, growing medium, and publication year, journal, and country, as well as challenges were retrieved from the selected studies. To properly respond to the RQs, all of the extracted data were categorized and synthesized into various classifications. The following sections present the results of this SLR.

## 3. Deep Learning in CEA


**RQ.1: What are the most often utilized DL models in CEA and their benefits and drawbacks?**


In CEA, DL models have been applied to a variety of tasks, such as crop phenotyping, disease and small insect detection, growth monitoring, nutrient status and stress level monitoring, microclimatic condition prediction, and robotic harvesting, all of which require large amounts of data for the machine to learn from. The architectures have been implemented in various ways, including deep belief network (DBN), convolutional neural network (CNN), recurrent neural networks (RNN), stacked auto-encoders, long short-term memory (LSTM), and hybrid approaches. CNN, which has three primary benefits including parameter sharing, sparse interactions, and equivalent representations, is a popular and commonly used approach in deep learning. CNN’s feature mapping includes *k* filters that have been spatially divided into several channels [102]. The feature map’s width and height are reduced using the pooling technique. CNNs use filters to capture the semantic correlations through convolution operations in multiple-dimensional data as well as pooling layers for scaling and shared weights for memory reduction to evaluate hidden patterns. As a result, the CNN architecture has a significant advantage in comprehending spatial data, and the network’s accuracy improves as the number of convolutional layers rises.

RNN and LSTM are very useful in processing time-series data, which are frequently utilized in CEA. The most well-known RNN variations include Neural Turing Machines (NTM), Gated Recurrent Units (GRU), and Long-Short Term Memory (LSTM), with LSTM being the most popular for CEA applications. Typically for data dimensionality reduction, compression, and fusion, autoencoders (AE) are used to automatically learn and represent the unlabeled input data. Encode and decode are two of the autoencoder’s operations. Encoding input images yields a code, which is subsequently decoded to get an output. The back-propagation technique is used to train the network so that the output is equal to the input. A DBN is created by stacking a number of distinct unsupervised networks, such as RBMs (restricted Boltzmann machines), so that each layer can be connected to both previous and subsequent layers. As a result, DBNs are often constructed by stacking two or more RBMs. It is significant to demonstrate that DBNs have been used in CEA applications [74]. The benefits and drawbacks of various DL models are listed in Table 5. Table 5 reveals that the identified drawbacks of DL methods prevent them from becoming canonical approaches in CEA. Each DL approach has the features that make it better suited than the others to a certain application in the CEA. Hybrid models are said to address the shortcomings of some of the single DL methods. The hybrid approach demonstrates the integration of several deep learning techniques. In the publications we reviewed, we discovered some studies that made use of the hybrid approach. Figure 4. shows a visual breakdown of the most often used CEA approaches along with how frequently they are applied.

The following subsection classifies CEA into two categories: (1) Greenhouse, (2) Indoor farm.

### 3.1. Deep Learning in Greenhouses


**RQ.2: What are the main application domains of DL in CEA?**


In this subsection, we present the DL models in greenhouse production for diverse applications. Table 3 present the application domain, tasks, DL model, network, optimizer, datasets, pre-processing augmentation, imaging method, growing medium and performance of DL in greenhouse.

#### 3.1.1. Microclimate Condition Prediction

Maintaining the greenhouse at its ideal operating conditions throughout all phases of plant growth requires an understanding of the microclimate and its characteristics. The greenhouse can increase crop yield by operating at the optimal temperature, humidity, carbon dioxide (CO2) concentrations, and other microclimate parameters at each stage of the plant growth. For instance, greater indoor air temperatures—which can be achieved by preserving the greenhouse effect or using the right heating technology—are necessary for the maximum plant growth in cold climates. On the other hand, the greenhouse effect is only necessary in very hot areas for a brief period of around 2–3 months while other suitable cooling systems are needed [103]. Accurate prediction of a greenhouse’s internal environmental factors using DL approaches is one of the recent trends in CEA. In our survey, we found 5 studies [30,31,32,33,34] that mentioned microclimate conditions prediction in the greenhouse.

#### 3.1.2. Yield Estimation

Crop detection, one of the most important topics in smart agriculture, especially in greenhouse production, is critical for matching crop supply and demand and crop management to boost productivity. Many of the surveyed articles demonstrate the application of DL models for crop yield estimation. The Single Shot MultiBox detector (SSD) method was used in the studies [37,43,51,53] to estimate tomato crops in the greenhouse environment followed by robotic harvesting. Other applications of SSD include detecting oyster mushrooms in [39] and sweet pepper in [49]. Another DL model called You Only Look Once (YOLO) with different modifications has been utilized in some of the reviewed papers for crop yield estimation as demonstrated in [36,41,46,47,51,52,53]. As described in [40,42,45,48,50,61], R-CNN models such as Mask-RCNN and Faster-RCNN, two of the most widely used DL models, are used in crop yield prediction applications, especially for tomato and strawberry. Other custom DL models for detecting crops have been proposed in the studies of [35,38,44,54].

#### 3.1.3. Disease Detection and Classification

Disease control in greenhouse environments is one of the most pressing issues in agriculture. Spraying pesticides/insecticides equally over the agricultural area is the most common disease control method. Although effective, this approach comes at a tremendous financial cost. Techniques for image recognition using DL can dramatically increase efficiency and speed while reducing recognition cost. As indicated in Table 3, we only identified various diseases of tomato and cucumber based on our assessments of the evaluated publications. As indicated in Table 3, we identified various diseases of tomato such as powdery mildew (PM) in [55,58,62], early blight in [55,58,63], leaf mold in [59,62,63], yellow leaf curl [59,63], gray mold in [62,63], spider mite in [60] and virus disease in [56]. Similarly, the diseases of cucumber such as powdery mildew (PM) in [55,57,58], downy mildew (DM) in [55,57,58,61] and virus disease in [58] are the sole diseases discussed based on our assessments of the evaluated publications. The wheat disease stated in [64] is another disease reported in the examined articles.

#### 3.1.4. Growth Monitoring

Plant growth monitoring is one of the applications where DL techniques have been applied to greenhouse production. Plant growth monitoring encompasses various areas such as length estimation at all crop growth stages as demonstrated in [76,77], and anomalies in plant growth in [78,82]. Other areas where plant growth monitoring is applied are in the prediction of Phyto-morphological descriptors as demonstrated in [79], seedling vigor rating in [80], leaf-shape estimation [83], and spike detection and segmentation in [81].

#### 3.1.5. Nutrient Detection and Estimation

It is crucial for crop management in greenhouses to accurately diagnose the nutritional state of crops because both an excess and a lack of nutrients can result in severe damage and decreased output. The goal of automatically identifying nutritional deficiencies is comparable to that of automatically recognizing diseases in that both involve finding the visual signs that characterize the disorder of concern. Based on our survey, we realized that there are few works dedicated to DL for nutrient estimation compared to most works utilizing DL for nutrient detection. The goal of nutritional detection is to identify one of these pertinent deficiencies, therefore symptoms that do not seem to be connected to the targeted disorders are disregarded. The studies [69,75] employed the autoencoders approach to detect nutrient deficiencies and lead content, respectively. CNN models were also frequently used in applications for nutrient detection. This was demonstrated in soybean leaf defoliation in [70], nutrient concentration in [72], nutrient deficiencies in [75], net photosynthesis modeling in [71] and calcium and magnesium deficiencies in [73]. As shown in [74], the cadmium concentration of lettuce leaves was estimated using a different DL model called DBN that was optimized using particle swarm optimization.

#### 3.1.6. Small Insect Detection

The intricate nature of pest control in greenhouses calls for a methodical approach to early and accurate pest detection. Using an automatic detection approach (i.e., DL) for small insects in a greenhouse is even more critical for quickly and efficiently obtaining trap counts. The most prevalent greenhouse insects discovered in the reviewed studies are whiteflies and thrips [65,66,67,68]. Our survey mentioned four studies for applying DL models (mostly CNN architectures) for tiny pest detection.

#### 3.1.7. Robotic Harvesting

Robotics has evolved into a new “agricultural tool” in an era where smart agriculture technology is so advanced. The development of agricultural robots has been hastened by the integration of digital tools, sensors, and control technologies, exhibiting tremendous potential and advantages in modern farming. These developments span from rapidly digitizing plants with precise, detailed temporal and spatial information to completing challenging nonlinear control tasks for robot navigation. High-value crops planted in CEA (i.e., tomato, sweet pepper, cucumber, and strawberry) ripen heterogeneously and require selective harvesting of only the ripe fruits. According to the reviewed papers, few works have utilized DL for robotic harvesting applications, such as picking-point positioning in grapes [85], obstacle separation using robots in tomato harvesting [84], 3D-pose detection for tomato bunch [86] and lastly, target tomato positioning estimation [87].

#### 3.1.8. Others

Other applications related to DL in CEA applications include predicting low-density polyethylene (LDPE) film life and mechanical properties in greenhouses using a hybrid model integrating both SVM and CNN [88].

### 3.2. Deep Learning in Indoor Farms

This subsection presents the main applications of the reviewed works that utilized DL in indoor farms (vertical farms, shipping containers, plant factories, etc.,). Table 4 present the application domain, tasks, DL model, network, optimizer, datasets, preprocessing augmentation, imaging method, growing medium, and performance of DL in indoor farms.

#### 3.2.1. Stress-Level Monitoring

To reduce both acute and chronic productivity loss, early detection of plant stress is crucial in CEA production. Rapid detection and decision-making are necessary when stress manifests in plants in order to manage the stress and prevent economic loss. We discovered that a few DL stress-level monitoring papers are reported for plant factories. Stress level monitoring encompasses various areas such as water stress classification [92], tip-burn stress detection [93], lettuce light stress grading [94], and abnormal leaves sorting [91].

#### 3.2.2. Growth Monitoring

In an indoor farm, it is critical to maintain a climate that promotes crop development through ongoing farm conditions monitoring. Crop states are critical for determining the optimal cultivation environment, and by continuously monitoring crop statuses, a proper crop-optimized farm environment can feasibly be maintained. In contrast to traditional methods, which is time-consuming, DL models are required to automate the monitoring system and increase measurement accuracy. We found several studies used DL models for growth monitoring in indoor farms, including plant biomass monitoring [99], growth prediction model in arabidopsis [97], growth prediction model in lettuce [95], vision based plants phenotyping [98], plant growth prediction algorithm [96,101] and the development of automatic plant factory control system [100].

#### 3.2.3. Yield Estimation

Due to its advantages over traditional methods in terms of accuracy, speed, robustness, and even resolving complicated agricultural scenarios, DL methods have been applied to yield estimation and counting research applications in indoor farming systems. The domains covered by yield estimation and counting from the examined publications include the identification of rapeseed [89] and cherry tomatoes [90].

The application distribution of DL techniques in CEA is shown in Figure 5.

## 4. Discussion

### 4.1. Summary of Reviewed Studies

We observed a rapid advancement in CEA using DL techniques between 2019 and 2022, as demonstrated in Figure 6. With rising work since 2019, this illustrates the relevance of DL in CEA. In Figure 7, we showed the distribution of published articles by various journals. The figure shows that the journal Computers and Electronics in Agriculture published the most DL for CEA articles (19). We also presented the country-by-country distribution of the evaluated articles, with China accounting for 40% of the total, indicating the highest number of publications, as shown in Figure 8. Korea and the Netherlands each contain 10% and 7% of the papers, respectively.

### 4.2. Evaluation Parameters

Our survey found that various evaluation parameters were employed in the selected publications **(RQ.3)**. Precision, recall, intersection-over-union (IoU), root mean square error (RMSE), mean average precision (mAP), F1-Score, root mean square error (RMSE), R-Square, peak signal noise ratio (PSNR), Jaccard index, success rate, sensitivity, specificity, accuracy, structural similarity index measure (SSIM), errors, standard error of prediction (SEP), and inference time were the most commonly used evaluation parameters for the DL analysis in CEA. Figure 9 depicts the frequency with which the assessment parameters are used. With 29 times, accuracy was the most frequently utilized as an evaluation measure. Precision, recall, mAP, F1-Score, and RMSE were used at least 10 times; IoU and R-Square were used 5 times, while the rest were used fewer than 5 times. We noticed that RMSE and R-Square were utilized as evaluation metrics in all microclimate prediction studies. Success rate and accuracy were used as evaluation measures for robotic harvesting applications. With the exception of a few cases of recall, precision, mAP, and F1-score, works related to growth monitoring applications used accuracy, RMSE, R-Square, and accuracy. RMSE, precision, recall, mAP, F1-Score, and accuracy were commonly utilized in other applications in the examined studies.

### 4.3. DL Backbone Networks


**RQ.4: What are the DL backbone networks used in CEA applications?**


There are many backbone networks, but this article will only focus on the backbone networks used in the reviewed papers, which include ResNet, EfficientNet, DarkNet, Xception, InceptionResNet, MobileNet, VGG, GoogleNet, PRPNet. These network structures are fine-tuned or combined with other backbone structures.

ResNet was the most often utilized network in CEA applications, according to the survey, as illustrated in Figure 10. The ResNet architecture can overcome the vanishing/exploding gradient problem [104]. When using gradient-based learning and backpropagation to train a deep neural network, the number of *n* hidden layers is multiplied by the *n* number of derivatives. The vanishing gradient problem occurs when the derivatives are modest, and the gradient rapidly diminishes as it spreads throughout the model until it vanishes. The gradient increases exponentially as the derivatives grow, resulting in the exploding gradient problem. A skip connection strategy is utilized in the ResNet to skip some training layers and connect directly to the output. The benefit of utilizing the skipping approach is that if any layer degrades the performance of the network, regularization will skip it, preventing exploding/vanishing gradient problems.

The main feature of MobileNet [105] is that it uses depth-wise separable convolutions to replace the standard convolutions of traditional network structures. Its significant advantages are high computational efficiency and small parameters of convolutional networks. MobileNet v1 and v2 are used in the reviewed articles, with v2 performing faster than v1. ResNet, on the other hand, adds a structure made up of multiple layers of networks that feature a shortcut connection known as a residual block. ResNet and FPN are used by Mask R-CNN to combine and extract multi-layer information. Many variants of ResNet architecture were discovered in reviewed articles, i.e., the same concept but with a different number of layers. A ResNeXt replicates a building block that combines a number of transformations with the same topology. It exposes a new dimension in comparison to ResNet, and requires minimal extra effort in designing each path.

Inception network [106] uses many tricks to push performance, both in terms of speed and accuracy, such as in dimension reduction. The versions of the inception network used in these reviewed papers are InceptionV2, InceptionV3, Inception-ResNetV2, and SSD InceptionV2. Each version is an upgrade to increase the accuracy and reduce the computational complexity. InceptionResNetV2 can achieve higher accuracies at a lower epoch. With the advantage of expanding network depth while using a small convolution filter size, VGG [107] can significantly boost model performance. VGGNet inherits some of its framework from AlexNet [108]. GoogleNet [109] has an inception module inspired by sparse matrices, which can be clustered into dense sub-matrices to boost computation speed, which is in contrast to AlexNet and VGGNet, which increases the network depth to improve training results. Contrary to VGG-nets, the Inception model family has shown that correctly constructed topologies can produce compelling accuracy with minimal theoretical complexity.

The backbone network for You Only Look Once (YOLO), DarkNet, has been enhanced in its most recent edition. YOLOv2 and YOLOv3 introduce DarkNet19 and DarkNet53, respectively, while YOLOv4 proposes CSPDarkNet [110]. CSPNet [111] is proposed to mitigate the problem of heavy inference computations from the network architecture perspective and has been seen to be used in the recent YOLO structure, i.e., SE-YOLOv5 [56]. Other backbone network structures include Xception [112] with different layers of 65 and 71, EfficientNet [113], and PRPNet [55].

### 4.4. Optimizer


**RQ.5: What are the optimization methods used for CEA applications?**


In contrast to the increasing complexity of neural network topologies [114], the training methods remain very straightforward. In order to make a neural network efficient, it must first be trained, as most neural networks produce random outputs without it. Optimizers, which modify the properties of the neural network, such as weights and learning rate, have long been recognized as a primordial component of DL, and a robust optimizer can dramatically increase the performance of a given architecture.

Stochastic gradient descent (SGD) is an optimization approach and one of the variants of gradient descent that is also commonly used in neural networks. It updates the parameters for each training one at a time, eliminating redundancy. As a hyper-parameter, the learning rate of SGD is often difficult to tune because the magnitudes of multiple parameters change greatly, and adjustment is required during the training process. Several adaptive gradient descent variants have been created to address this problem, including Adaptive Moment Estimation (Adam) [115], RMSprop [116], Ranger [117], Momentum [118], and Nesterov [119]. These algorithms automatically adapt the learning rate to different parameters, based on the statistics of gradient leading to faster convergence, simplifying learning strategies, and have been seen in many neural networks applied to CEA applications, as demonstrated in Figure 11.

### 4.5. Growing Medium and Plant Distribution


**RQ.6: What are the primary growing media and plants used for DL in the CEA?**


We note that the most common growing medium used in the evaluated studies is soil-based (78%), as shown in Figure 12. There are 14 publications on hydroponics, one on aquaponics, and none on aeroponics for soil-less growing media. This insinuates that these soilless growing media are still in their infancy. We also showed the distribution of the plants used in the evaluated papers, with tomatoes representing 39% of all plants grown in the CEA and corresponding to the highest number of publications, as shown in Figure 13. The percentages of papers that planted lettuce, pepper, and cucumber are 16%, 9%, and 8%, respectively. According to the reviewed publications, it was also discovered that indoor farms used soil-less techniques (hydroponics and aquaponics) more frequently than greenhouse systems, which frequently used soil-based growing medium.

### 4.6. Challenges and Future Directions

To the best of our knowledge, the paragraphs below provide a brief description of some specific aspects on the challenges and potential directions of DL applications in CEA.

For DL models to be effective, learning typically needs a lot of data. Such huge training datasets are difficult to gather, not publicly available for some CEA applications, and may even be problematic owing to privacy laws. Even while data augmentation and massive training datasets methods can somewhat make up for the shortage of huge labeled datasets, it is difficult to completely meet the demand for hundreds or thousands, if not less, high-quality data points. When utilized with validated data, DL models may not be able to generalize in situations where the data is insufficient. However, we discovered a number of studies that used smaller datasets and attained great accuracy, as shown in [40,45,56,59,82]. The studies demonstrated various strategies for handling this circumstance by carefully choosing the features that ensure the method will perform at its peak. Additionally, in order to ensure optimal performance and streamline the processing of the learning algorithms, the dimensionality of the input vectors for the classification and detection algorithms must be reduced.

DL algorithms are also susceptible to the caliber of the data utilized to train them. Overfitting can occur when an algorithm “learns” about noise and excessive details in the input set, which has a detrimental effect on the created model’s ability to generalize. The model in this instance performs admirably on the training dataset but poorly on new data. To combat the overfitting model, regularization techniques include weight decay/regularization, altering the network’s complexity (i.e., the amount of weights and their values), early halting, and activity regularization.

We expect in the future to see more combinations of two-time series models for temporal sequence processing as demonstrated in [31]. It is also anticipated that more methods would use LSTM or other RNN models in the future, utilizing the time dimension to make more accurate predictions, especially in climatic condition prediction.Additionally, it helps to gauge the reliability of time series prediction by offering an explicable result. As a result, improving interpretability will receive a lot of attention in the future [120].

The majority of the evaluated studies focused on supervised learning, while just a small number used semi-supervised learning. Future works that include unsupervised learning into CEA applications will be heavily reliant on tools like the generative adversarial network (GAN). A generative modeling method known as GAN learns to replicate a specific data distribution. The lack of data is a major barrier to creating effective deep neural network models, but GANs are the solution [121]. In order to lessen model overfitting, the realistic images created by GAN that differ from the original training data are appealing in data augmentation of DL-computer vision.

Another area worth noting is the clear interest in the use of AI and computer vision in CEA applications. With the use of DL-computer vision, a number of difficult CEA issues are being resolved. However, DL-computer vision does face significant difficulties, one of which is the enormous processing power. Adopting cloud-based solutions with auto scaling, load balancing, and high availability characteristics is one way to deal with this issue. Real-time video input analysis and real-time inferences are some of the limitations of cloud solutions, but edge devices with features like GPU accelerators can do it. Utilizing computer vision solutions on edge hardware helps lessen latency restrictions. Few works have addressed the need for proper security to ensure data integrity and dependability in the rapidly expanding field of computer vision in CEA; additional research into this area is needed in subsequent works.

There is an imperative need where deep learning needs to be applied in the next few years such as developing more microclimate models for monitoring and maintaining the microclimatic parameters to the desired range for optimal plant growth and development, thus helping in irrigation and fertigation management of the crops. The need for AI, particularly DL, to derive an empirical and non-linear “growth response function” that maps microclimate conditions to crop growth stages is critical because, according to the reviewed papers, this has not been extensively studied. This calls for the optimization of microclimate control set points at various growth stages of crops. There are currently very few publications that have developed prediction models for the microclimate parameters in CEA. In addition to the microclimate prediction models, the need to also develop more microclimate control systems such as (1) developing automatic shading system to prevent crops from harsh sunlight in greenhouses, (2) developing pad-fan systems and fogging systems based on vapor pressure deficit (VPD) control, which is an effective way to simultaneously maintain ideal ranges of temperature and relative humidity, thus significantly enhancing plant photosynthesis and productivity in greenhouse production, (3) developing photoperiod control systems based on light spectrum and intensity control. Despite the paucity of studies on microclimate prediction and control, extensive research is needed in the use of edge-AI systems for precise monitoring at various phases of crop growth. Lastly, it is crucial to investigate the use of DL for nutrient solution management in soilless cultures (influenced by both microclimate conditions and crop growth). We anticipate that further research that considers monitoring, predicting, controlling, and optimizing microclimate factors in CEA will become available in the near future as advancements in accuracy, efficiency, and architectures are put forth. Additionally, the labor availability and associated costs, are a growing concern for the sustainability and profitability of CEA industry. Some research has been reported for developing robotic systems, but majority of it is focused on field production. However, the CEA is a unique production environment and the indoor grown crops have different requirements for automation based on the production technology employed (greenhouse, vertical tower, vertical tier, hydroponic, dutch bucket, pot/tray, etc., ). Further, the CEA crops are more dense (plants per unit area), which makes robotics applications more challenging. Thus, extensive efforts are required to develop DL-driven automation and robotic systems for different production environments, to address these challenges.

## 5. Conclusions

Today, it is evident that prediction and optimization procedures are essential in many industries. This study has fully discussed a review of DL-based research efforts in CEA, which were motivated by the most recent breakthroughs in computational neuroscience. This study examined various application areas, described the tasks, listed technical details such as DL models and networks, described the preprocessing augmentation, the optimizer used, and performance of each method.

The results of this study demonstrate that the applications of DL models have attracted a lot of interest recently as a result of their ability to recognize distinctive object features and offer greater precision. There is no way to determine which DL model is the best. However, we found that RNN-LSTM was frequently used for predicting microclimate conditions in CEA due to its time series prediction. We noticed that prediction of the microclimate conditions, a crucial issue in CEA, was the subject of relatively little of the reported research. We can see that CNN models, the widely used DL model, have high applicability and universality based on the reviewed papers. CNN and ResNet are most widely adopted DL model and network, while other models and networks are also implemented in this domain. In order to generate constructive discussions of the limitations of DL techniques in the CEA domain, critical challenges and future research prospects were presented. We believe these studies will serve as a roadmap for future studies towards creating an intelligent system for various CEA applications.

## Figures and Tables

**Figure 1 sensors-22-07965-f001:**
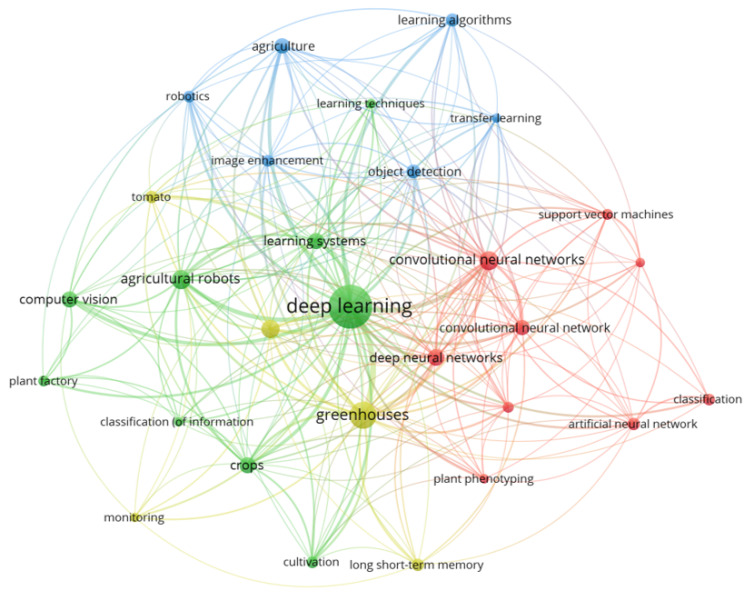
Bibliometric visualization produced by VOSviewer Software using the author’s specified keywords.

**Figure 2 sensors-22-07965-f002:**
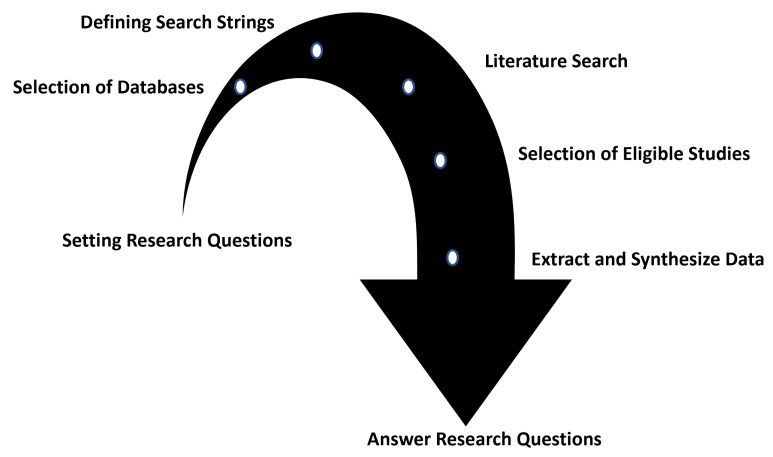
Planning and reporting process of systematic literature review (SLR).

**Figure 3 sensors-22-07965-f003:**
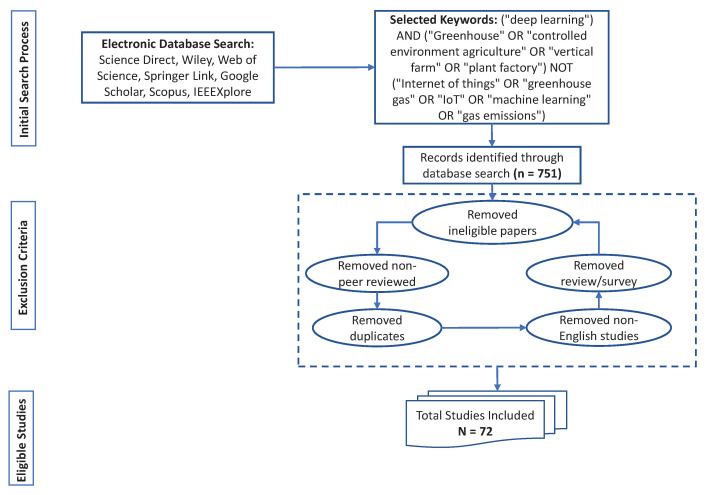
Article inclusion and exclusion process flowchart.

**Figure 4 sensors-22-07965-f004:**
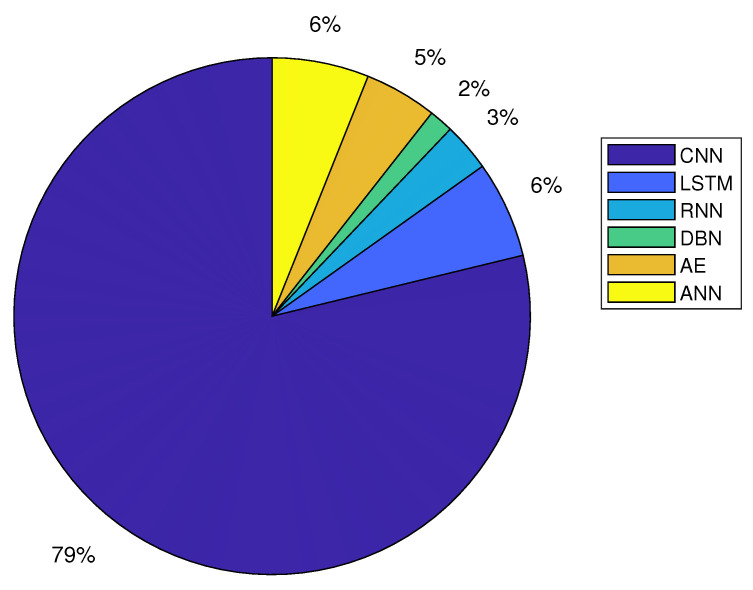
Visual illustration of the deep learning techniques applied to controlled environment agriculture in 2019–2022 (Focusing on the reviewed papers).

**Figure 5 sensors-22-07965-f005:**
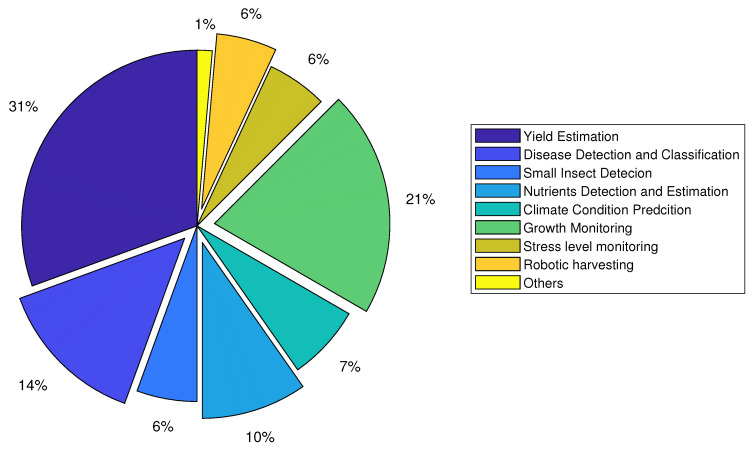
Application distribution of deep learning in controlled environment agriculture.

**Figure 6 sensors-22-07965-f006:**
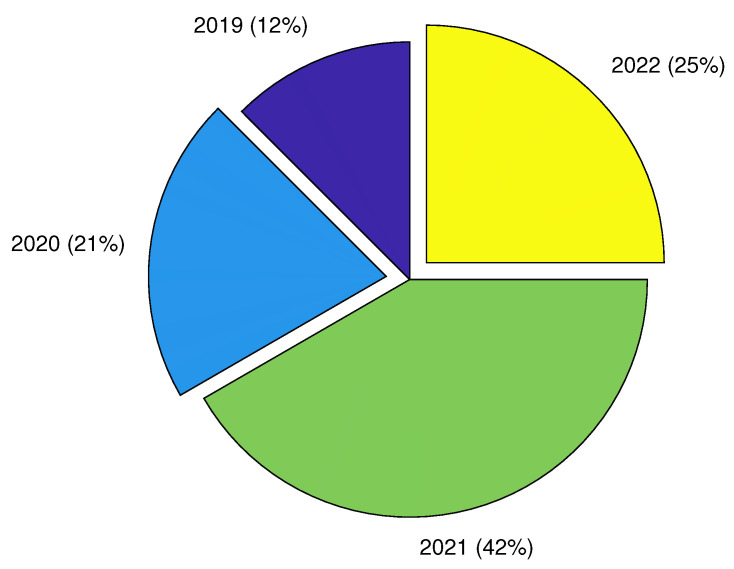
Year-wise distribution of the publication from 2019 to April 2022.

**Figure 7 sensors-22-07965-f007:**
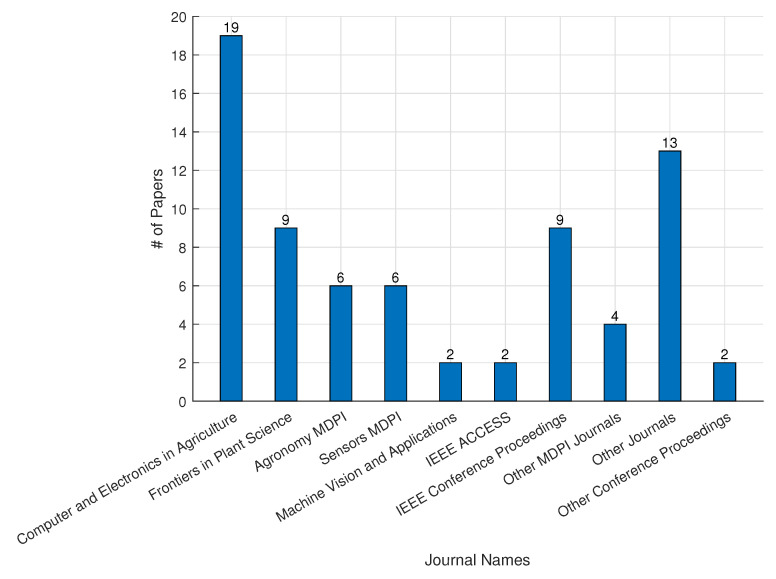
Publication distribution for deep learning applications in controlled environment agriculture.

**Figure 8 sensors-22-07965-f008:**
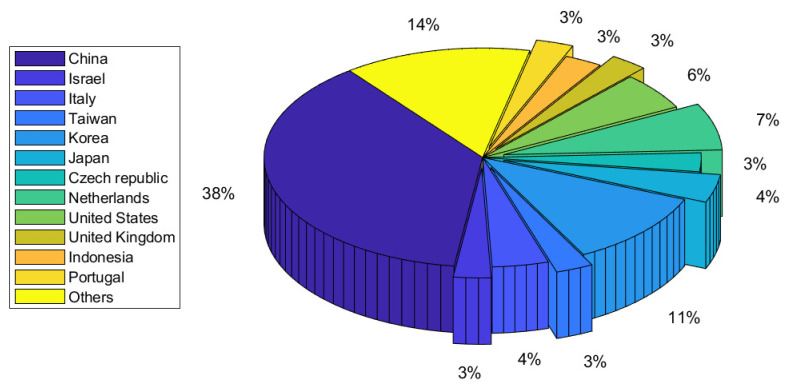
Country-wise distribution of the reviewed papers in controlled environment agriculture.

**Figure 9 sensors-22-07965-f009:**
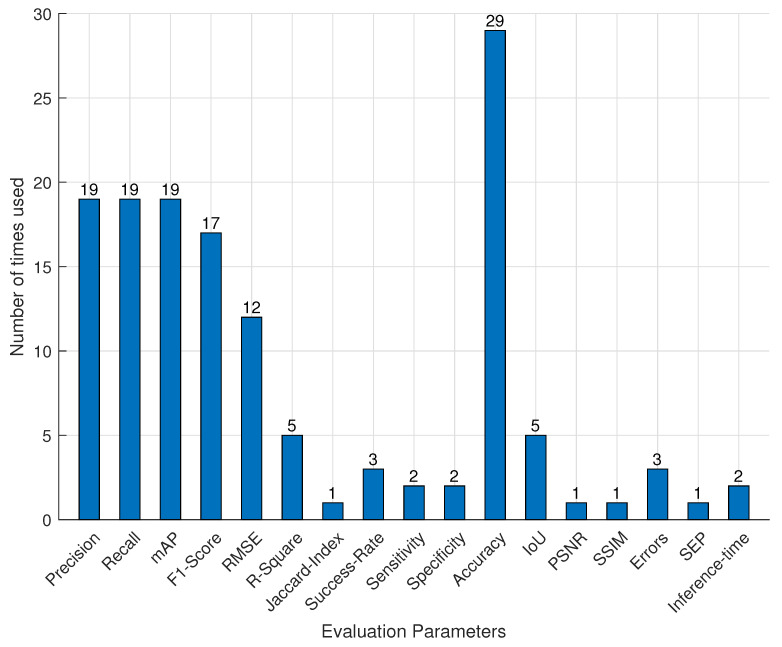
Evaluation parameters distribution of deep learning model in controlled environment agriculture.

**Figure 10 sensors-22-07965-f010:**
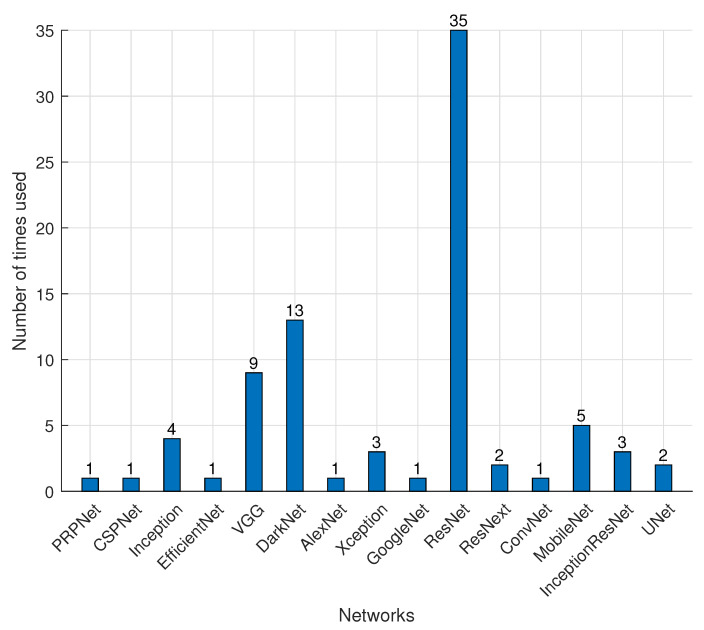
Distribution of different deep learning training networks used in controlled environment agriculture.

**Figure 11 sensors-22-07965-f011:**
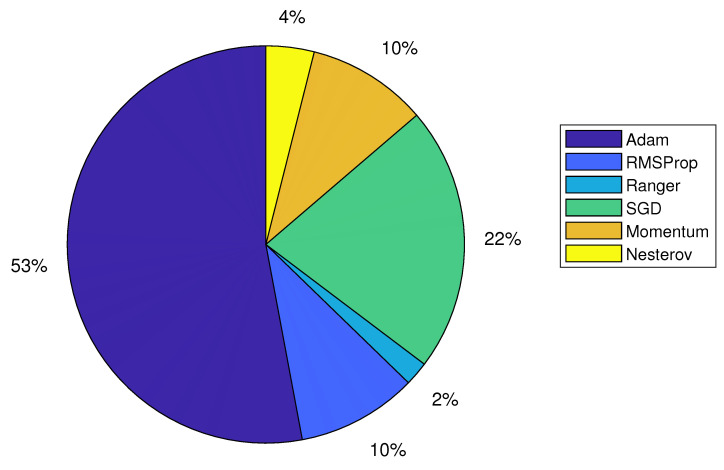
Distribution of different deep learning optimizer used in controlled environment agriculture.

**Figure 12 sensors-22-07965-f012:**
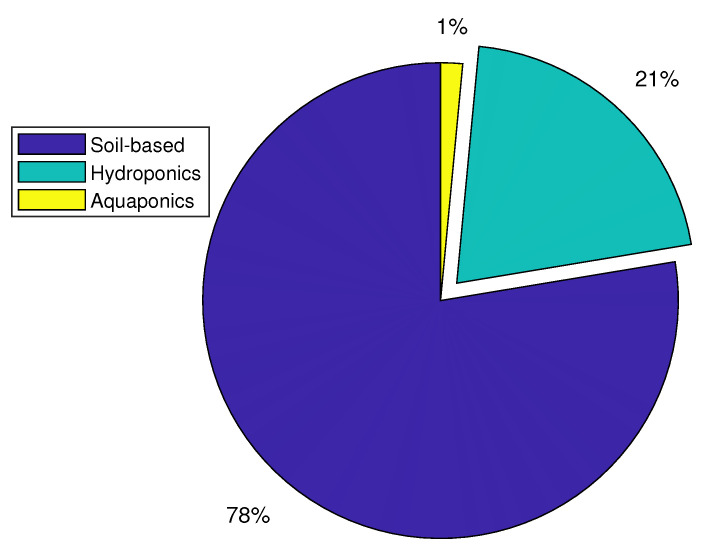
Growing medium distribution in controlled environment agriculture.

**Figure 13 sensors-22-07965-f013:**
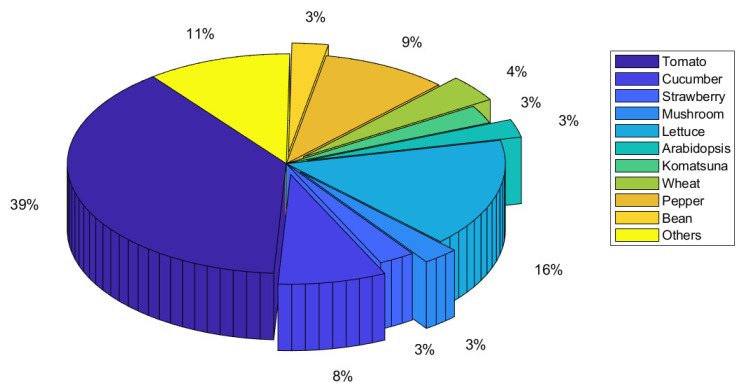
Plant distribution of papers for deep learning applications in controlled environment agriculture.

**Table 1 sensors-22-07965-t001:** Summary of the recent important related reviews.

Ref.	Year	Focus of Study	Highlights
[22]	2018	Deep learning in agriculture	40 papers were identified and examined in the context of deep learning in the agricultural domain.
[23]	2019	Fruit detection and yield estimation	The development of various deep learning models in fruit detection and localization to support tree crop load estimation was reviewed.
[24]	2019	Plant disease detection and classification	A thorough analysis of deep learning models used to visualize various plant diseases was reviewed.
[25]	2020	Dense images analysis	Review deep learning applications for dense agricultural scenes, including recognition and classification, detection counting, and yield estimation.
[26]	2021	Plant disease detection and classification	Current trends and limitations for detecting plant leaf disease using deep learning and cutting-edge imaging techniques.
[27]	2021	Weed detection	70 existing deep learning-based weed detection and classification techniques cover four main producers: data acquisition, datasets preparation, DL techniques, and evaluation metrics approaches.
[28]	2021	Bloom/Yield recognition	Diverse automation approaches with computer vision and deep learning models for crop yield detection were presented.
Our Paper	2022	Deep learning applications in CEA	Review developments of deep learning models for various applications in CEA.

**Table 2 sensors-22-07965-t002:** Distribution of papers selected from different databases.

Source	Number of Papers in the Initial Search	Eligible Papers with Duplicates
Google Scholar	330	27
Scopus	127	25
Science Direct	119	19
Wiley	40	4
IEEEXplore	51	9
SpringerLink	44	4
Web of Science	40	17
**Total**	**751**	**105**

**Table 3 sensors-22-07965-t003:** Summary of studies for deep learning applications in greenhouses.

Application Classification	Tasks	Growing Medium	DL Model	Networks	Preprocessing Augmentation	Optimizer	Dataset Type	Imaging Method	Performance	Ref.
Climate Condition Prediction	Transpiration rate	hydroponic	ANN	ANN	NS	Adam	31,033 data points	NS	RMSE = 0.07–0.10-gm${}^{-2}$ min${}^{-1}$, R2 = 0.95–0.96	[30]
	temp (°C), humidity deficit (g/kg), relative humidity (%), radiation $\left(W/m^{2}\right)$, ${\mathrm{CO}}_{2}$ conc.	soil-based	RNN-TCN	LSTM-RNN	NS	Adam	NS	NS	RMSE (Dataset1): 10.45(±0.94), RMSE (Dataset2): 6.76 (±0.45), RMSE(Dataset3): 7.40 (±1.88)	[31]
	temperature, humidity, ${\mathrm{CO}}_{2}$ concentration	soil-based	ANN	NS	NS	Adam	NS	NS	ANN at 30 min, R2 = (temp: 0.94, humidity: 0.78, ${\mathrm{CO}}_{2}$: 0.70), RMSEP = (temp: 0.94, humidity: 5.44, ${\mathrm{CO}}_{2}$: 32.12), %SEP = (temp: 4.22, humidity: 8.18, ${\mathrm{CO}}_{2}$: 6.49)	[32]
			NARX						NARX at 30 min, R2 = (temp: 0.86, humidity: 0.71, ${\mathrm{CO}}_{2}$: 0.81), RMSEP = (temp: 1.32, humidity: 6.27, ${\mathrm{CO}}_{2}$: 28.30), %SEP = (temp: 5.86, humidity: 9.42, ${\mathrm{CO}}_{2}$: 7.74)	
			RNN-LSTM						RNN-LSTM at 30 min, R2 = (temp: 0.96, humidity: 0.8, ${\mathrm{CO}}_{2}$: 0.81), RMSEP = (temp: 0.71, humidity: 5.23, ${\mathrm{CO}}_{2}$: 28.30), %SEP = (temp: 3.15, humidity: 7.85, ${\mathrm{CO}}_{2}$: 5.72)	
	temp., humidity, pressure, dew point	soil-based	RNN-LSTM	NS	NS	NS	NS	NS	Temperature, RMSE = 0.067163	[33]
	temp., humidity, illumination, ${\mathrm{CO}}_{2}$ conc., soil temp. and soil moisture	soil-based	LSTM	NS	NS	NS	NS	NS	Temp., RMSE = 0.38 (tomato), 0.55 (cucumber), 0.42 (pepper)	[34]
									Humidity, RMSE = 1.25 (tomato), 1.95 (cucumber), 1.78 (pepper)	
									Illumination, RMSE = 78 (tomato), 80 (cucumber), 30 (pepper)	
									${\mathrm{CO}}_{2}$ , RMSE = 3.2 (tomato), 4.1 (cucumber), 3.9 (pepper)	
									Soil temp., RMSE = 0.07 (tomato), 0.08 (cucumber), 0.045 (pepper)	
									Soil moisture, RMSE = 0.14 (tomato), 0.30 (cucumber), 0.15 (pepper)	
Yield Estimation	corn crop and leaf weeds classification	soil-based	Dual PSPNet	ResNet-50	rotation, shift (height, width, vertical, horizontal, pixel intensity), zoom and Gaussian blur	SGD with Nesterov Momentum	6906 images	RGB	Balanced Accuracy (BAC) = 75.76%, Dice-Sorensen Coefficient (DSC) = 47.97% (for dataset A+C)	[35]
	green pepper detection	soil-based	Improved YOLOv4-tiny	CSP DarkNet53	Gaussian noise addition, HSL adjustment, scaling and rotation	NS	1500 images	RGB	P: 96.91%, R: 93.85%, AP: 95.11%, F1 Score: 0.95	[36]
	cherry tomato clusters location detection, tomato’s maturity estimation	soil-based	SSD	MobileNet V1	horizontal flip and random crop	Adam or RMSprop	254 images	RGB	IoU = 0.892 (for tomato’s cluster location detection), RMSE: 0.2522 (for tomato’s maturity estimation)	[37]
	tomato organs detection	soil-based	Improved FPN	ResNet-101	NS	SGD	8929 images	RGB	mAP: 99.5%	[38]
	mushroom recognition	soil-based	Improved SSD	MobileNet V2	flip, random rotation, random cropping, and random size, brightness and tone conversion, random erasure, mixup	NS	4600 images	RGB	P: 94.4%, R: 93%, mAP: 93.2%, F1 Score: 0.937, Speed: 0.0032s	[39]
	tomato detection	soil-based	Mask R-CNN	ResNext-101	NS	SGD	123 images	RGB	P: 93%, R: 93%, F1 Score: 0.93	[40]
	mushroom localization	soil-based	YOLOv3	DarkNet53	NS	NS	500 images	RGB	Average prediction error = 3.7 h, Average detection = 46.6	[41]
	tomato detection	hydroponic	Faster R-CNN	ResNet-101	gamma correction	momentum	895 images	RGB, HSV	detection accuracy: 88.6%	[42]
	cherry tomato detection	soil-based	SSD	MobileNet	rotating, brightness adjustment and noising	RMSProp	1730 images	RGB	AP = 97.98%	[43]
				InceptionV2					AP = 98.85%	
				SSD300		Adam			AP = 92.73%	
				SSD512					AP = 93.87%	
	plant classification	soil-based	The LNet270v1	custom	random reflection (X and Y), Shear (X and Y), Scale (X and Y), Translation (X and Y), rotation	Adam	13,766 images	RGB	mean accuracy: 91.99%, mIoU: 86.5%, mean BFScore: 86.42%	[44]
	tomato detection	soil-based	Mask R-CNN	ResNet-50	None used	SGD	123 images	RGB	Average result @ 0.5, (ResNet-50, P = 84.5%, R = 90.5%, F1 Score = 0.87)	[45]
				ResNet-101					Average result = 0.5, (ResNet-101, P = 82.5%, R = 90%, F1 Score = 0.86)	
				ResNext-101					Average result @ 0.5, (ResNext-101, P = 92%, R = 93%, F1 Score = 0.925)	
	Lettuce seedlings identification	hydroponic	YOLO-VOLO-LS	VOLO	rotation, flipping, and contrast adjustment	NS	6900 images	RGB	Average results = (recall: 96.059%, Precision: 96.014%, F1-score: 0.96039)	[46]
	Fig detection	soil-based	YOLOFig	ResNet43	NS	NS	412 images	RGB	P = 74%, R = 88%, F1-score = 0.80	[47]
	strawberry detection	soil-based	Improved Faster-RCNN	ResNet-50	brightness, chroma, contrast, and sharpness augmentation and attenuation	NS	400 images	RGB	Accuracy = 86%, ART = 0.158s, IoU = 0.892	[48]
	sweet pepper detection	soil-based	SSD	custom	NS	NS	468 images	RGB, HSV	Average Precision = (Flash-only: 84%, Flash-No-Flash image: 83.6%)	[49]
	tomato detection	soil-based	Faster R-CNN	ResNet-50, ResNet-101, Inception-ResNet-v2	resizing, crop, rotating, random horizontal flip	NS	640 images	RGB	F1 score = 83.67% and AP = 87.83% for tomato detection using Faster R-CNN with ResNet-101, R2 = 0.87 for tomato counting	[50]
	tomato detection	soil-based	SSD	MobileNetv2	rotation, translate, flip, multipley, noise addition, scale, blur	NS	1029 images	RGB, HSV	mAP = 65.38%, P = 70.12%, R = 84.9%, F1-score = 85.81%	[51]
			YOLOv4	CSP DarkNet53					mAP = 65.38%, P = 70.12%, R = 84.9%, F1-score = 85.81%	
	muskmelon detection	soil-based	YOLO Muskmelon	ResNet43	NS	NS	410 images	RGB	IoU = 70.9%, P = 85%, R = 82%, AP = 89.6%, F1 = 84%, FPS = 96.3	[52]
	tomato detection	soil-based	SSD	MobileNet V2	rotation, scaling, translation, flip, blur (Gaussian Filter), Gaussian Noise	NS	5365	RGB	mAP = 51.56%, P = 84.37%, R = 54.40%, F1 = 66.15%, I = 16.44 ms	[53]
				InceptionV2					mAP = 48.54%, P = 85.31%, R = 50.93%, F1 = 63.78%, I = 24.75 ms	
				ResNet-50					mAP = 42.62%, P = 92.51%, R = 43.59%, F1 = 59.26%, I = 47.78 ms	
				ResNet-101					mAP = 36.32%, P = 88.63%, R = 38.13%, F1 = 53.32%, I = 59.78 ms	
			YOLOv4-tiny	CSP DarkNet53					mAP = 47.48%, P = 88.39%, R = 49.33%, F1 = 63.32%, I = 4.87 ms	
	Arabidopsis, Bean, Komatsuna recognition	soil-based	CNN	ResNet-18	scaling, rotation and translation	Adam	2694 images	RGB	mA = 0.922 (Arabidopsis), mA = 1 (Bean), mA =1 (Komatsuna)	[54]
Disease Detection and Classification	Tomato (powdery mildew (PM), early blight) and cucumber (PM, downy mildew (DM)) recognition	soil-based	CNN	PRP-Net	ShiftScaleRotate, RandomSizedCrop, HorizontalFlip	SGD	4284 images	RGB	Average results (Accuracy = 98.26%, Precision = 92.60%, Sensitivity = 93.60%, Specificity = 99.01%)	[55]
	tomato virus disease recognition	soil-based	SE-YOLOv5	CSPNet	Gaussian noise addition, rotation, mirroring, intensity random adjustment	NS	150 images	RGB, HSV	P = 86.75%, R = 92.19%, mAP@(0.5) = 94.1%, mAP@(0.5:0.95) = 75.98, prediction accuracy = 91.07%	[56]
	cucumber PM, DM and the combination of PM and DM recognition	soil-based	Efficient Net	EfficientNet-B4	flip (horizontal, vertical), rotation	Ranger	2816 images	RGB	Train Accuracy = 99.22%, Verification accuracy = 96.38%, Test accuracy = 96.39%	[57]
	Tomato (PM, early blight), cucumber (PM, DM, virus disease) recognition	soil-based	ITC-Net	ResNet18 and TextRCNN	Cropping, Normalization, word segmentation, word list construction, text vectorization	Adam	1516 images	RGB	Accuracy: 99.48%, Precision: 98.90%, Sensitivity: 98.78%, Specificity: 99.66%	[58]
	leaf mold, tomato yellow leaf curl detection	soil-based	CNN	ResNet-50, ResNet-101	filtering, histogram	NS	115 images	RGB, HSV	Testing Accuracy = 98.61%, Validation accuracy = 99%	[59]
	spider mite detection	soil-based	CNN	ResNet18	NS	NS	850 images	multi-spectral, RGB	accuracy: 90%	[60]
	cucumber DM prediction	soil-based	LSTM	NS	Min-Max normalization	Adam	11,827 images	RGB	A = 90%, R = 89%, P = 94%, F1-Score = 0.91	[61]
	tomato disease detection	soil-based	Faster R-CNN	VGG16	resizing, cropping, rotation, flipping, contrast, brightness, color, noise	NS	59,717 images	RGB	mAP = 89.04%	[62]
				ResNet-50					mAP = 90.19%	
				ResNet-50 FPN					mAP = 92.58%	
	various tomato diseases (i.e., leaf mold, gray mold, early blight, late blight, leaf curl virus, brown spot) detection	soil-based	YOLO-Dense	DarkNet53	NS	NS	15,000 images	RGB	mAP: 96.41%	[63]
	wheat disease detection	soil-based	CNN	ResNet-101	cropping	NS	160 plants	NIR, RGB	Accuracy = 84% for tan spot disease, 75% for leaf rust disease	[64]
Small Insect Detection	Pests (whitefly and Thrips) detection	soil-based	TPest-RCNN	VGG16	Resizing, Spliting	NS	1941 images	RGB	AP: 95.2%, F1 Score: 0.944	[65]
	whiteflies (greenhouse whitefly and cotton whitefly) detection	hydroponic	Faster R-CNN	ResNet-50	mirroring	SGD	1161 images	RGB	RMSE = 5.83, Precision = 0.5794, Recall = 0.7892	[66]
	whitefly detection	soil-based	YOLOV4	CSP DarkNet53	cropping	Adam	1200 images	RGB	Whitefly: (precision = 97.4%, recall = 95.7%), mAP = 95.1%	[67]
	Thrips detection								Thrips: (precision = 97.9%, recall = 94.5%), mAP = 95.1%	
	flies, gnats, thrips, whiteflies detection	soil-based	YOLOv3-tiny	DarkNet53	cropping	Adam	NS	RGB	average F1-score: 0.92, mean counting accuracy: 0.91	[68]
Nutrient Estimation and Detection	lead content detection	soil-based	WT-MC-stacked auto-encoders	NS	standard normalized variable (SNV), 1st Der, 2nd Der, 3rd Der, 4th Der	NS	2800 images	hyper-spectral data	pb content detection = 0.067∼1.400 mg/kg, RMSEC = 0.02321 mg/kg, RMSEP = 0.04017mg/kg, R2C = 0.9802, R2P = 0.9467	[69]
	soyabean leaf defoliation estimation	soil-based	CNN	AlexNet	Resizing, Binarized, Rotation	NS	10,000 images	RGB	RMSE (AlexNet) = 4.57(±5.8)	[70]
				VGGNet					RMSE (VGGNet): 4.65 (±6.4)	
				ResNet					RMSE (ResNet): 14.60 (±18.8)	
	PN: (light level ${\mathrm{CO}}_{2}$ concentration, temperature) prediction	soil-based	DNN	custom	NS	Adam	33,000 images	NS	accuracy: 96.20% (7 hidden layer with 128 units per hidden layer), accuracy: 96.30% (8 hidden layer with 64 units per hidden layer)	[71]
	nutrient concentration estimation	hydroponic	CNN	VGG16	width, height shift, shear, flipping, zoom, scaling, cropping	Adam	779 images	RGB	Average Classification Accuracy (ACA) = 97.9%	[72]
				VGG19					Average Classification Accuracy (ACA) = 97.8%	
	Calcium Magnesium deficiencies prediction	soil-based	SVM, Random Forest (RF) Classifier	Inception V3	NS	RMSProp	880 images	RGB	Accuracy = 98.71% (for InceptionV3 with SVM) and 97.85% (for Inception-V3 with RF classifier)	[73]
				VGG16		Adam			Accuracy = 99.14% (for VGG16 with SVM) and 95.71% (for VGG16 with RF classifier)	
				ResNet-50		Adam			Accuracy = 88.84% (for ResNet50 with SVM) and 84.12% (for ResNet-50 with RF classifier)	
	cadmium content estimation	soil-based	PSO-DBN	NS	Savitzky-Golay(SG) to remove the spectral noise	NS	1260 images	hyper-spectral data	When the hidden layers is 3, the prediction result is as follows, R2: 0.8976, RMSE: 0.6890, and RPD: 2.8367	[74]
	Nutrient deficiencies (Calcium/Ca2+, Potassium/K+, Nitrogen/N) classification	soil-based	CNN	Inception-ResNetV2	shift, rotation, resizing	NS	571 images	RGB	Average Accuracy = 87.27%, Average Precision = 100%, Recall = Ca2+: 100%, K+: 100%, N: 100%	[75]
			Auto-Encoder	NS					Average Accuracy = 79.09%, Average Precision = 94.2%, Recall = Ca2+: 97.6%, K+: 92.45%, N: 95.23%	
Growth Monitoring	length estimation and interest point detection	soil-based	Mask R-CNN	ResNet-101	NS	NS	2574 images	RGB	Results in 2D (Banana Tree, AP: 92.5%, Banana Leaves, AP: 90%, Cucumber fruit, AP: 60.2%)	[76]
	internode length detection	soil-based	YOLOv3	DarkNet53	NS	NS	9990 images	RGB	R:92% AP: 95%, F1 Score: 0.94	[77]
	plant growth anomalies detection	soil-based	LSTM	NS	filtering, cropping	Adam	NS	RGB, HSV	2D (P: 42% R: 71%, F1: 0.52), 3D photogrammetry with high resolution camera (P: 57% R: 57%, F1: 0.57), 3D low-cost photogrammetry system (P: 44% R: 79%, F1 :0.56), LiDAR (P: 5% R: 86%, F1: 0.63)	[78]
	Phytomorphological descriptor prediction	aquaponics	CNN	DarkNet53	Scaling and Resizing	SGD with Momentum	300 images	RGB	R2(Area-DarkNet53) = 0.9858, R2(Diameter-DarkNet53) = 0.9836	[79]
				Xception					R2(Centroid x-Xception) = 0.6390, R2(Centroid-y-Xception) = 0.7239	
				Inception ResNetv2					R2(Major Axis-InceptionResNetv2) = 0.8197, R2(Minor Axis-InceptionResNetv2) = 0.7460	
	orchid seedlings vigor rating	soil-based	CNN	ResNet-50	Cropping, Resizing	Adam	1700 images	RGB, HSV	A = 95.5%, R = 97%, P = 94.17%, F1-Score = 0.9557	[80]
	spike detection	soil-based	SSD	Inception-ResNetv2	NS	SGD	292 images	RGB	AP@0.5 = 0.780, AP@0.75 = 0.551, AP@0.5:0.95 = 0.470	[81]
			YOLOv3	DarkNet53	NS				AP@0.5 = 0.941, AP@0.75 = 0.680, AP@0.5:0.95 = 0.604	
			YOLOv4	CSP DarkNet53	CutOut, MixUp, CutMix, RandomErase				AP@0.5 = 0.941, AP@0.75 = 0.700, AP@0.5:0.95 = 0.610	
			Faster R-CNN	InceptionV2	NS	Adam			AP@0.5 = 0.950, AP@0.75 = 0.822, AP@0.5:0.95 = 0.660	
	spike segmentation		ANN	NS	NS	NS			AP = 0.61	
			U-Net	VGG16	rotation [−30 30], horizontal flip, and brightness	Adam			AP = 0.84	
			Deep-LabV3+	ResNet-101		NS			AP = 0.922	
	Paprika leaves growth conditions classification	soil-based	DNN	Improved VGG-16	rotation	NS	227 images	hyper-spectral data	Accuracy = 90.9%	[82]
				VGG-16					Accuracy = 86.4%	
				ConvNet					Accuracy = 82.3%	
	leaf shape estimation	hydroponic	encoder-decoder CNNs	U-Net	random rotation, and random horizontal spatial flipping	Adam	NS	RGB	Deviation of U-Net based estimation is less than 10% of the manual LAI estimation	[83]
Robotic Harvesting	Obstacle Separation	soil-based	Mask R-CNN	ResNet-101	3D HSI color thresholding	NS	NS	RGB	Success Rate = 65.1% (whole process)	[84]
	picking-point positioning	soil-based	CNN	custom	NS	NS	100 images	RGB	Success rate: 100%	[85]
	keypoints detection	soil-based	TPM	custom	Rotation and brightness adjustment	RMSprop	2500 images	RGB	Qualified rate: 94.02%, Accuracy: 85.77%	[86]
	pose detection					Adam			Accuracy: 70.05%	
	target positioning estimation	soil-based	Mask-RCNN	ResNet	cropping	NS	NS	RGB, Infrared	Average Gripping Accuracy (AGA): 8.21mm, APSR: 73.04%	[87]
Others	LPDE film lifetime prediction	NS	SVM-CNN	NS	NS	Adam	4072 images	NS	NS	[88]

NS: Not Specified.

**Table 4 sensors-22-07965-t004:** Summary of studies for deep learning applications in indoor farms.

Application Classification	Tasks	Growing Medium	DL Model	Networks	Preprocessing Augmentation	Optimizer	Dataset Type	Imaging Method	Performance	Ref.
Yield Estimation	rapeseed detection	hydroponic	ESPA-YOLO-V5s	CSP DarkNet	rotating, flipping (horizontal, vertical)	NS	6616 images	RGB	P = 94.5%, R = 99.6%, F1-score = 0.970, mAP@0.5 = 0.996	[89]
	tomato prediction	hydroponic	Improved Mask R-CNN	ResNet	random translation, random brightness change, Gaussian noise addition	NS	1078 images	RGB	Accuracy = 93.91% (Fruit), Accuracy = 88.13% (Stem)	[90]
Stress Level Monitoring	lettuce abnormal leaves (yellow, withered, decay)	hydroponic	DeepLabV3+	Xception-65	rotating, mirroring, flipping	NS	500 images	RGB	Xception-65 (mIoU = 0.4803, PA = 95.10%, speed = 243.4 ± 4.8a)	[91]
				Xception-71					Xception-71 (mIoU = 0.7894, PA = 99.06%, speed = 248.9 ± 4.1a)	
				ResNet-50					ResNet-50 (mIoU = 0.7998, PA = 99.20%, speed = 154.0 ± 3.8c)	
				ResNet-101					ResNet-101 (mIoU = 0.8326, PA = 99.24%, speed = 193.4 ± 4.0b)	
	water stress classification	NS	CNN	ResNet50	rotation, re-scaling	SGD with momentum /Adam /RMSProp	800 images	RGB	Average Accuracy: ResNet-50 with (Adam = 94.15%, RMSProp =88.75%, SGDm = 83.77%)	[92]
				GoogLeNet					GoogLeNet with (Adam = 78.3%, RMSProp = 80.4%)	
	patch-level detection	NS	YOLOv2	DarkNet19	NS	SGD with Nesterov Momentum	60,000 images	RGB	Accuracy = 87.05%	[93]
	pixel-level segmentation		U-Net	NS	cropping, random jittering	Adam			mAP = 87.00%, IoU = 77.20%, Dice score = 75.02%	
	light stress grading	hydroponic	MFC-CNN	custom	90, 180, and 270-degree rotation, mirror rotation, salt and pepper noise, and image sharpening	SGD	1113 images	RGB	Accuracy = 87.95% Average F1-score = 0.8925	[94]
Growth Monitoring	plant growth prediction	NS	NS	NS	NS	NS	45 data samples	RGB	RMSE = 0.987, R2 = 0.728 for 4-7-1 network architecture	[95]
	leaf shape estimation	NS	custom	Spatial transformer network	rotation, scaling, translation	Adam	NS	RGB	PSNR = 30.61, SSIM = 0.8431	[96]
									PSNR = 26.55, SSIM = 0.9065	
									PSNR = 23.03, SSIM = 0.8154	
	growth prediction	soil-based	U-Net	SE-ResXt101	cropping, scaling and padding	NS	232 plant samples	RGB	F1-score = 97%	[97]
	plant behaviour prediction	hydroponic	Mask R-CNN	NS	rotation and scaling	NS	1728 images	RGB	leaf area accuracy = 100%	[98]
	lettuce plant biomass prediction	hydroponic	DCNN	ResNet-50	rotation, brightness, contrast, saturation, hue, grayscale	Adam	864 plants	RGB	For RGBD (MAPE = 7.3%, RMSE = 1.13g), For RGB (MAPE = 9.6%, RMSE = 1.03g), For Depth (MAPE = 12.4%, RMSE = 2.04g)	[99]
	growth prediction	hydroponic	ANN	NS	NS	NS	NS	NS	ANN: Accuracy (%) = 98.3235, F-measure (%) = 97.5413, Training time (sec) = 121.78	[100]
			SVM						SVM: Accuracy (%) = 96.0886, F-measure(%) = 93.4589, Training time (sec) = 202.48	
	growth prediction	hydroponic	Mask R-CNN	ResNet-50	flipping, cropping and rotation	NS	600 images	NS	mAP = 76.9%, AP = 92.6%	[101]

**Table 5 sensors-22-07965-t005:** Common DL architectures with their benefits and drawbacks.

Model	Ref.	Advantages	Disadvantages
AE	[69,75]	Excellent performance for depth feature extractions Do not need labeled data for training Saves a significant amount of time by avoiding labeling in the case of large datasets	Lengthy processing time and fine tuning Training may be hampered by errors that vanishes
DBN	[74]	Unsupervised training High efficiency in handling hyperspectral data at high dimensions Can simplify characteristics that are redundant and complex through training network layer by layer	Disable to process multi-dimensional Training can be prolonged and inefficient
LSTM	[31,61]	Able to capture abstract temporal features Alleviate the diminishing gradient problems	Poor spatial features representation resulting in classification errors Difficult implementation
ANN	[30,32]	Excellent for obtaining significant findings from complex nonlinear data Can make highly accurate approximations of a vast class of functions. Quite robust to noise in the training data.	Weak stability in heavily interconnected and complex systems Require many training sets
CNN	[45,62]	Ability to learn robust discriminative features Ability to capture spatial correlations High generalization potential	High computational cost Difficult parameter tuning

## Data Availability

Not applicable.

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
