# Peer review of "Deep Learning in Controlled Environment Agriculture: A Review of Recent Advancements, Challenges and Prospects"

_sensors, 2022, doi:10.3390/s22207965_

Round 1

Reviewer 1 Report

Authors have prepared an article related to Deep Learning in Controlled Environment Agriculture. The article is neatly written. Basically, the article is about bibliometric analysis rather than the survey article.  I would suggest following points to authors,

1. In the “Table 2. Distribution of papers selected from different databases”, you have considered paper sources as Scopus. Generally, Scopus covers papers published in other publisher sources such as Science Direct, Wiley, IEEEXplore, SpringerLink. How did you eliminate the repeated articles during the article search process? What was the criteria for considering unwanted articles?

2. Though the article is largely inclined towards bibliometric analysis, explain the need of bibliometric analysis for Deep Learning in Controlled Environment Agriculture. Highlight the importance.

3. In the “Figure3: Article inclusion and exclusion process flowchart”, include the number of papers in all the intermediate steps as well.

Author Response

Response to the Decision on Manuscript “Deep Learning in Controlled Environment Agriculture: A

Review of Recent Advancements, Challenges and Prospects”

We would like to thank you for your time and effort. All the concerns have been carefully addressed. We sincerely hope that our revised manuscript is now in satisfactory form. Below, we have provided answers to your comments.

Review 1.1

Authors have prepared an article related to Deep Learning in Controlled Environment Agriculture. The article is neatly written. Basically, the article is about bibliometric analysis rather than the survey article.  I would suggest following points to authors,

  1. In “Table 2. Distribution of papers selected from different databases”, you have considered paper sources as Scopus. Generally, Scopus covers papers published in other publisher sources such as Science Direct, Wiley, IEEEXplore, SpringerLink. How did you eliminate the repeated articles during the article search process? What was the criteria for considering unwanted articles?

Response 1.1

The authors are thankful for your comment. We searched different databases so that we can cover all the literature related to DL in Controlled Environment Agriculture. We sort it out with the use of Mendeley, as well as sorting it out manually.

Review 1.2

Though the article is largely inclined towards bibliometric analysis, explain the need of bibliometric analysis for Deep Learning in Controlled Environment Agriculture. Highlight the importance.

Response 1.2

The authors are thankful for your comment. There are few papers that have reviewed deep learning in agriculture, however, there is no paper that has surveyed or reviewed several papers related to Deep learning in controlled environment agriculture (CEA). CEA is a growing industry in the agriculture sector, and the increasing popularity of DL applications in CEA sparks our motivation to conduct a systematic review of recent advances in this domain.

Review 1.3

In the “Figure3: Article inclusion and exclusion process flowchart”, include the number of papers in all the intermediate steps as well.

Response 1.3

The authors are thankful for your comment. Since we have explained how we sorted out the articles both with Mendeley and manually, we feel there is no need to modify figure 3. Adding intermediate steps are trivial for the scope of the article.

Reviewer 2 Report

In this paper, the recent advancements, challenges and prospects of DL in CEA were  reviewed. It has important value to promote this research and application. The paper was well organized with clear review purpose, method and conclusion.

As the decision of DL model may not only focus on the application, but also be related to the crop features and CEA envirionment conditions,  it is suggested to increase some discussion about the challenges and prospects. 

Author Response

Response to the Decision on Manuscript “Deep Learning in Controlled Environment Agriculture: A

Review of Recent Advancements, Challenges and Prospects”

We would like to thank you for your time and effort. All the concerns have been carefully addressed. We sincerely hope that our revised manuscript is now in satisfactory form. Below, we have provided answers to your comments

Review 2.1

In this paper, the recent advancements, challenges and prospects of DL in CEA were  reviewed. It has important value to promote this research and application. The paper was well organized with clear review purpose, method and conclusion.

As the decision of DL model may not only focus on the application, but also be related to the crop features and CEA envirionment conditions,  it is suggested to increase some discussion about the challenges and prospects. 

Response 2.1

The comments are appreciated by the authors. The section under "Challenges and Future Directions" now includes more pertinent discussion. Please take note of the blue-font highlighted text in the updated paper (line 464 -505).

Reviewer 3 Report

Deep Learning in Controlled Environment Agriculture: A Review of Recent Advancements, Challenges and Prospects

The current work presents a thorough review of the application of DL in CEA. It is an interesting subject that may provide support for future works in the field. One topic that I believe that could be added is the use of computer vision with DL in CEA. I would suggest the authors to either add a short section, or at least a few paragraphs in the discussion with possible combinations of CV applied to CEA in the future, in the conclusion. There are several recent works of CV using different devices (from cameras to smartphones) applied to classification of agricultural products, soil analysis, vegetation and leaf quality, etc, that I believe would be useful in the future on topics related to this review. I hereby list a few to provide support for the authors, but they may feel free to find others:

-    -    Cocoa Companion: Deep Learning-Based Smartphone Application for Cocoa Disease Detection - 10.1016/j.procs.2022.07.013

-    Deep learning based computer vision approaches for smart agricultural applications - https://doi.org/10.1016/j.aiia.2022.09.007

-    Deep-learning-based in-field citrus fruit detection and tracking - https://doi.org/10.1093/hr/uhac003

I have only minor comments for the authors.

I believe that section 2 extends the text without relevant information to the main subject, as it reports the methods in the research. I would suggest the authors to present it as supplementary material, or else synthetize or remove it from the main text.

Author Response

Response to the Decision on Manuscript “Deep Learning in Controlled Environment Agriculture: A

Review of Recent Advancements, Challenges and Prospects”

We would like to thank you for your time and effort. All the concerns have been carefully addressed. We sincerely hope that our revised manuscript is now in satisfactory form. Below, we have provided answers to your comments.

Review 3.1

Deep Learning in Controlled Environment Agriculture: A Review of Recent Advancements, Challenges and Prospects

The current work presents a thorough review of the application of DL in CEA. It is an interesting subject that may provide support for future works in the field. One topic that I believe that could be added is the use of computer vision with DL in CEA. I would suggest the authors to either add a short section, or at least a few paragraphs in the discussion with possible combinations of CV applied to CEA in the future, in the conclusion. There are several recent works of CV using different devices (from cameras to smartphones) applied to classification of agricultural products, soil analysis, vegetation and leaf quality, etc, that I believe would be useful in the future on topics related to this review. I hereby list a few to provide support for the authors, but they may feel free to find others:

-    -    Cocoa Companion: Deep Learning-Based Smartphone Application for Cocoa Disease Detection - 10.1016/j.procs.2022.07.013

-    Deep learning based computer vision approaches for smart agricultural applications - https://doi.org/10.1016/j.aiia.2022.09.007

-    Deep-learning-based in-field citrus fruit detection and tracking - https://doi.org/10.1093/hr/uhac003

Response 3.1

The comments are appreciated by the authors. The section under "Challenges and Future Directions" now includes more pertinent discussion. Please take note of the blue-font highlighted text in the updated paper (line 464 -505).

Review 3.2

I have only minor comments for the authors.

I believe that section 2 extends the text without relevant information to the main subject, as it reports the methods in the research. I would suggest the authors to present it as supplementary material, or else synthetize or remove it from the main text.

Response 3.2

The authors are thankful for the comments. The research methodology is needed to provide a guide for the review study on how to systematically refine the search results to narrow it down to a specific topic of interest.